# Role Stress, Job Burnout, and Job Performance in Construction Project Managers: The Moderating Role of Career Calling

**DOI:** 10.3390/ijerph16132394

**Published:** 2019-07-05

**Authors:** Guangdong Wu, Zhibin Hu, Junwei Zheng

**Affiliations:** 1School of Public Affairs, Chongqing University, Chongqing 400044, China; 2Department of Construction Management, Jiangxi University of Finance & Economics, Nanchang 330013, China; 3Department of Construction Management, Kunming University of Science and Technology, Kunming 650500, China

**Keywords:** role stress, job burnout, career calling, job performance

## Abstract

This study aims to explore the influence of role stress (role ambiguity and role conflict) on job burnout and job performance in construction project managers in the Chinese construction industry. Based on the JD-R (Job Demands Resources) model, this study introduces career calling as the moderating variable, in order to develop a theoretical model. The theoretical model is then tested with structural equation modeling. This work uses data from 191 owners, contractors, subcontractors, and supervisors in the Chinese construction industry. The results indicate that: (i) role ambiguity has a negative and significant effect on job burnout and job performance; (ii) role conflict has a negative effect on job burnout, but has a non-significant influence on job performance; (iii) job burnout has a negative impact on job performance; (iv) career calling negatively moderates the relationship between role ambiguity and job burnout, and positively moderates the relationship between role conflict and job performance. Furthermore, the results also show that career calling can positively moderate the effect of role conflict on job burnout. This study expands the existing body of knowledge by reasonably controlling role stress and appropriately introducing career calling. In addition, the study provides some suggestions relevant to construction project management.

## 1. Introduction

Construction is characterized as having complex, high-working demands within its internal and external environments. It is a dynamic, risky and hazardous industry [1]. The Chinese construction industry has been expanding at a high growth rate over the past 20 years. In order to alleviate the effects of the global financial crisis, the Chinese government has vowed to increase investments in housing and infrastructural works to stimulate the national economy [2]. In 2017, China’s construction industry grew rapidly, with its total output value reaching 21,395 billion yuan, an increase of 10.5% over the previous year, much higher than 7.1% in 2016 and 2.3% in 2015. This is equivalent to 25.9% of China’s Gross Domestic Product (GDP), while the total added value accounts for 6.7% of GDP. The contribution of the construction industry to China’s economy is also evident. China’s construction contractors are undertaking a large number of domestic and foreign construction projects, bringing huge occupational pressure and psychological burden to construction project managers (CPMs) [3]. Busy and committed construction project managers (CPMs) have to face common traditional challenges such as cost, time and quality, and they must also cope with ever-pressing safety and environmental issues [4]. In practice, CPMs co-ordinate the activities of every project participant to ensure they complete their intended tasks within an appropriate time frame, thus contributing to a more efficacious project team [5]. In this case, it is inevitable that CPMs are subject to a great deal of role stress in their work. Job burnout has been suggested as a major reaction to and product of job stress [6], which can adversely affect the CPM’s job performance. As such, CPMs suffer from high psychological stress and strong turnover intention. In order to alleviate the role stress and job burnout of CPMs and improve their job performance, most researchers focus on the effect of organizational support or social support (e.g., [7,8]). However, simply relying on organizational support or commitment is not sufficient to moderate role stress and job burnout. This is because construction projects have temporary characteristics, which makes it difficult to provide long-term and effective support for CPMs. A CPM is also required to exert their subjective initiative to adjust their emotions and work enthusiasm [9], such as career calling. Career calling, as a public service motivation, is a component of job resources that contributes to alleviating job burnout and role stress. Thus, more and more scholars are beginning to pay attention to the application of career calling in organizational management (e.g., Refs. [10,11,12]). In addition, career calling, from the individual subjective experience, reflects the basic goals of work and the importance of work in life, which will also influence job emotions and behaviors [13]. However, although the scale of construction projects continues to expand and are gradually tending to become larger, more complicated and integrated [14], few studies to date have explored the challenges faced by the career calling of CPMs and career calling’s effect on regulating job burnout and improving job performance.

The fact is that few studies have explored the effects of role stress on job burnout and job performance among CPMs, especially with regard to examining career calling as a moderating variable. This study aims to explore how role stress in CPMs and job burnout affects job performance, based on the JD-R model. In the JD-R model, role stress and job burnout belong to the category of job demands, which keeps individuals in a state of fatigue and leads to physical and psychological exhaustion. Career calling, as a subjective initiative, is one component of job resources that contributes to the alleviation of job burnout and role stress. This study provides significant theoretical and practical insights into stress management and job burnout management, as well as providing a reliable reference on how to enhance CPMs’ job performance and achieve project success.

## 2. Theoretical Background

### 2.1. JD-R Model

The JD-R model is a mainstream conceptual framework used in job stress and job burnout research. The model describes the influence of job characteristics on job burnout. The JD-R model was established to overcome some of the limitations that characterize earlier research models in the field of work psychology, including the job demand-control (JD-C) model [15], the job demand-control-support (JD-C-S) model [16], and the effort reward imbalance (ERI) model [17]. One of the drawbacks of these earlier models is their focus upon the source of independent, single, short-term job stressors, such as task overload, work overtime and task difficulties [7]. However, job burnout is induced not only by task-related factors but also by individual, occupational, organizational, and other social factors [18]. According to the JD-R model, job characteristics can be aggregated into two broad categories: job demands and job resources. Job demands refer to the material, psychological, social or organizational requirements involved in a person’s work that require continuous physical or psychological effort or skill [19], which in turn is related to certain physiological and psychological costs, such as job stress [20] and role ambiguity [21]. The term “job resources” refers to the material, psychological, social or organizational resources that help individuals achieve their work goals, reduce work demands and stimulate individual growth, learning and development [22]. Based on the JD-R model, researchers can better explain and predict individual job burnout. They can also integrate job engagement and well-being into the research framework, in order to carry out various horizontal and vertical empirical studies [18]. Due to the fact that construction projects involve many stakeholders who complete a number of complex tasks within a limited period of time [23], CPMs, who play a key role in achieving project success, often experience a great deal of stress. It is this stress, in turn, which puts CPMs at high risk of job burnout [24]. In the construction industry, long working hours, cruel market competition and society’s extra expectations of CPMs are all factors that reflect just some of CPMs’ job demands. When job demands continue to increase over a long period of time, CPMs need to make continuous efforts or take appropriate measures and strategies to actively respond to those demands. These measures and strategies reflect the positive role of job resources. Moreover, the JD-R model has been widely used to study stress and job burnout in professionals in construction projects and to indicate the interactive relationship between job demands and job resources, as well as their effect on job performance [5,23,24]. For these reasons, this study adopts the JD-R model to conduct theoretical analyses and construct a hypothetical model to investigate the relationship between role stress, job burnout and job performance among CPMs in China.

### 2.2. Role Stress

The stress people feel at work is called job stress, and role stress is an important component of job stress. Kahn [25] first defined role stress as the pressure that individuals face when they are unable to learn or understand the relevant rights and obligations related to their work and to perform their roles well. Subsequently, Hardy [26] believes that role stress is an imbalance status caused by some external factors. It is these factors which disturb the internal stability and cause individuals to be unable to express themselves with appropriate role behaviors in the social structure. There are two main perspectives on the dimension of role stress. One view holds that role stress can be classified into two dimensions: role conflict and role ambiguity [27,28]. Through an in-depth study of role stress, another view deems that role stress can be divided into three dimensions, namely role conflict, role ambiguity and role overload [25,29]. Role overload, however, is not studied as frequently as the first two [30]. In addition, Rizzo and Van Sell believed that a person may play many roles and may need to exhibit various behaviors in order to satisfy their position in the organization system [28,31]. As such, role overload can actually be attributed to an individual’s internal role conflict, which is essentially a role conflict. Additionally, many researches recognize a two-component nature of job-related role stress, consisting of role ambiguity and role conflict [32,33]. Hence, this study divides role stress into two structural dimensions: role conflict and role ambiguity. Role conflict means that, when individuals are faced with two or more role expectations, they cannot meet both of these expectations at the same time [34]. Role ambiguity, on the other hand, refers to employees’ feelings when they are unclear or lack proper understanding of their role and cannot obtain clear role expectations at work [35]. When faced with tasks and information conveyed by multiple role requirements, CPMs cannot determine how they can possibly balance the requirements of different roles, thus causing role conflict [27]. Similarly, CPMs will lack a clear understanding of their roles, thus leading to role ambiguity, when they are confused about their roles because of internal or external factors [36]. Construction has always been regarded as an industry with high levels of job stress [37]. The role of a CPM includes not only planning, organizing, and supervising the project team, but also handling time pressures, internal uncertainties, and the dynamic stakeholders inevitably found in construction projects [23]. Under the vested interest demands of multiple stakeholders, if a CPM does not deal with issues well, decision-making mistakes will occur, causing the CPM to doubt their own ability, causing even greater role conflict and role ambiguity.

### 2.3. Job Burnout

Job burnout refers to the negative feelings experienced by individuals in the work environment. Examples include physical and mental exhaustion, reduced work achievement, and reduced enthusiasm for work [38]. In recent years, the dimension and measurement of job burnout have mainly been based on the theoretical model of Maslach [39]. She proposes that job burnout can be analyzed from three dimensions: emotional exhaustion, cynicism and low professional efficacy. Emotional exhaustion refers to the emotional exhaustion caused by work and an unwillingness to invest any further emotion in work. Cynicism means querying the motives of others, generating negative emotions and treating them with a numb, ruthless attitude. Low professional efficacy refers to a negative evaluation of one’s own work that does not bring any practical influence to others. Because a construction project is a typical representative of a mission-driven industry, and one which involves many stakeholders, the participants need to complete many complicated and varied tasks in a limited time [40]. As such, CPMs are responsible for planning, organization and coordination throughout the entire construction process [41]. They face multiple pressures from tasks and interpersonal relationships, which can easily lead to negative feelings, such as emotional exhaustion, cynicism, low professional efficacy, and even increased job burnout. Pinto [42] also points out that the level of risk and job burnout faced by CPMs is much higher than that faced by managers in other industries, due to their reduced achievement and enthusiasm from an overloaded job. Therefore, it is very necessary to study the job burnout of CPMs, with a view to thereby reducing their stress, improving their work enthusiasm, and promoting the smooth completion and delivery of construction projects.

### 2.4. Job Performance

In order to achieve continuous improvement, employee performance must be regularly evaluated and monitored [43]. Deadrick [44] defines job performance as the achieved work outcomes for each job function during a specified period of time. El-Sabaa [45] explains that job performance is related to the willingness and openness to try and achieve new aspects of the job, which will bring about an increase in the individual’s productivity. Through the years, the dimension of job performance has been a primary topic of empirical investigation in applied psychology and management areas. Early on, Katz [46] divided job performance into two dimensions, namely extra-role behavior and in-role behavior. Subsequently, Borman [47] classified job performance into task performance and contextual performance. Task performance is defined as in-role behavior, which means that employees achieve the expected goal of the organization by adopting the behavior prescribed by the organization. Contextual performance is an extra-role behavior, which refers to activities that show special enthusiasm, during the course of work, to complete tasks, cooperate with others, and actively perform tasks above and beyond organizational requirements [48,49]. However, construction is a task-oriented industry whose main purpose is to meet the requirements of cost, time and quality [50]. Cheng [51] suggests that task performance is significantly related to final project outcomes. Therefore, contextual performance is conducive to achieving organizational goals. It is not a task activity and does not involve direct production or service activities. Therefore, in this study, job performance is defined as the behavior of CPMs who strive to achieve the established organizational standards and preset project objectives together with the organizational goals. Job performance is also based on the complexity and uncertainty of the construction project and the project team attributes. Based on the above analysis, this study considers task performance as the sole criterion for CPM job performance.

### 2.5. Career Calling

The sense of calling originated from religion and has the meaning of inspiration. With the development of China’s social economy, the religious meaning of career calling has gradually faded, while its common meaning is booming rapidly. Bellah [52] first introduced career calling into the workplace as a mission in work orientation. In later research on career calling, scholars held heated discussions and put forward different viewpoints. Hall [11] interprets career calling as a job that individuals feel is meaningful. This feeling can help individuals to achieve their own goals and is the purpose of individuals in the world. In addition, Dobrow Riza [53] maintains that career calling comes from the heart of the individual, not just the work itself, but is also the individual’s subjective perception of meaning. Furthermore, Elangovan [12] emphasizes that career calling comes from both external and internal sources and mainly includes three elements: (1) action orientation, (2) clarity of purpose and (3) mission and pro-social intention. Therefore, career calling is an incentivizing force that encourages people to contribute and commit to an organization, and which will more likely help the individual to find meaning in their job [54]. However, existing studies mainly focus on public professional roles, such as teachers [55], police [56] and other professions. Few studies focus on the career calling of CPMs. In fact, CPMs play an irreplaceable role in construction project networks, especially in complex construction projects. Construction projects have temporary characteristics, making it difficult to provide long-term and effective support for CPMs. Therefore, a CPM is required to exert their own career calling, to adjust their emotions and behavior, and adapt to changes in the work environment. Career calling, as a public service motivation, is a type of job resource that contributes to alleviating negative employee outcomes [9]. In the process of a CPM participating in management, the CPM’s career calling will be significantly different from other industry personnel. A CPM’s career calling brings deeper dedication and organizational responsibility, which is more conducive to career planning, making correct decisions, and achieving project success [57].

## 3. Hypotheses Development and Theoretical Model

### 3.1. Hypotheses Development

#### 3.1.1. Role Stress and Job Burnout

Roles in construction projects are diverse and include a wide range of issues which come about. Some of these issues come about because of the way the market operates, some because of the need for governance, while some have become institutionalized [58]. As construction tasks continue to evolve and become more complex, role stress becomes more complicated and tends to have a negative impact on work commitments. Role conflict, as a component of role stress, may cause negative emotional experiences for personnel, including increased tension, severe internal conflicts, decreased job satisfaction, and reduced confidence [25]. In addition, role conflict makes it impossible for CPMs to take into account the problems faced by others filling various roles. This role conflict will hinder the CPM’s identification with the organization and willingness to work hard. If role conflict involves serious conflict between role expectations and a CPM’s own values, it will also reduce their willingness to stay and increase the likelihood of job burnout intention [59].

When a CPM is ambiguous about his or her various roles—whether due to their own or external factors—they lack a clear understanding of their roles, work content and purpose. Therefore, the ambiguity of their roles will lead to the uncertainty of their own work [36]. This uncertainty makes employees constantly spend their own resources on seeking and obtaining related information. When employees do not receive additional resources, there will be an imbalance between resources and needs, which in turn can easily lead to job dissatisfaction and job burnout [18,60]. Therefore, role stress causes a CPM to lack the required, exact understanding of project goals and responsibilities. This leads to the phenomenon in which they are at a loss in the organization. Due to the existence of conflict and ambiguity in a CPM’s understanding of the organization, it is difficult for a CPM to clearly identify with the values and corporate culture of the organization, to find the right direction for their efforts, and to establish their loyalty to the organization. As such, their willingness to stay will be impaired [61]. Therefore, the following hypotheses are developed:
**Hypothesis** **1a** **(H1a):**Higher levels of role conflict will lead to higher levels of job burnout.
**Hypothesis** **1b** **(H1b):**Higher levels of role ambiguity will lead to higher levels of job burnout.

#### 3.1.2. Job Burnout and Job Performance

Research suggests that job burnout is associated with negative outcomes for individuals. Job burnout has, for example, been associated with health problems, anxiety, depression, reduced self-esteem, and substance abuse [24,62]. For the relevant industry, burnout has been shown to have a variety of dysfunctional consequences, including turnover, absenteeism, and reduced performance on the job, all resulting in significant costs to the individual and organization [63,64]. Burke [65] believes that job burnout is also a sequential process. That is, an individual experiencing job burnout will also have a negative effect on his or her colleagues. Due to the fact that the construction industry is a typical task-driven industry, a CPM’s job is typically characterized by “role overload, frenetic activity, and superficiality,” due to the wide scope of their responsibilities coupled with limited resources and authority [66]. Therefore, CPMs experience a significant level of job burnout because of an endless list of demands, deadlines, and problems throughout the project’s life cycle [67]. All of these factors can easily lead to lower levels of organizational effectiveness, job satisfaction and organizational commitment [68]. Therefore, the following hypothesis is developed:
**Hypothesis** **2a** **(H2a):**Higher levels of job burnout will lead to lower levels of job performance.

#### 3.1.3. Role Stress and Job Performance

Different scholars expounded on the outcome variables of role stress from different perspectives. Among them, job performance is the most frequently studied variable. Some scholars believe that role stress has a negative impact on job performance. For instance, Fisher [69], through meta-analysis, finds a negative correlation between role conflict and job performance. Moreover, Örtqvis [70] maintains that, when the expectation of the role sender to the role receiver is contradictory, the receiver will be confused and thus unable to better arrange his own work. This will ultimately reduce job performance.

Role ambiguity, as the lack of clear and available information for an expected role and position, is negatively correlated with job performance [71]. Furthermore, Jackson [72] believes that the cause of the impact of role ambiguity on job performance is that, due to either a lack of information or an excessive amount of information, the role recipients cannot obtain relevant behavior knowledge. This negatively affects the individual’s ability to work and reduces the level of job performance. However, Bhuian and Nygaard [73,74] believe that human beings actually need some stress to create the satisfaction that comes from achievement and to thwart feelings of boredom. According to this view, moderate levels of stress are beneficial, whereas low and high levels of stress undermine an individual’s wellbeing and performance [75,76].

Construction projects involve stakeholders from multiple organizational disciplines and functional areas [77]. A CPM’s job requires constant interactive engagement with stakeholders. However, interacting with multiple organizations and multiple tasks may confuse project managers with regard to their roles in achieving different project goals. This could also affect their judgment and decision-making ability, thereby influencing their job performance. Therefore, the following hypotheses are developed:
**Hypothesis** **3a** **(H3a):**Role conflict is related to job performance.
**Hypothesis** **3b** **(H3b):**Role ambiguity is related to job performance.

#### 3.1.4. The Moderating Role of Career Calling

Career calling, an incentivizing force that may come from external sources or from individuals themselves, is a person’s sense and understanding of the meaning and purpose of their work [78]. Moreover, career calling is also often pro-social in nature, meaning that viewing one’s career as a calling includes a desire to make the world a better place [79]. Hence, many scholars believe that career calling has a positive effect on job engagement. Hall [11] advanced the idea that individuals with a strong career calling tend to set clear goals and strive to achieve career success. Such goals may include the improvement of their own abilities. More interestingly, people with career calling are more inclined to find challenging jobs [80]. For organizations, employees with a strong career calling are more willing to invest greater effort and engagement [81], so they can also achieve better job performance [55], experience higher job satisfaction [82], and have a higher sense of attachment to the organization (e.g., clearer goals and low turnover) [10]. In addition, career calling not only reduces individual role stress [83], job burnout, and maintains mental health; it also improves an employee’s work meaning to obtain life satisfaction and experience happiness [84]. In reality, CPMs co-ordinate the activities of every project participant, to ensure they complete their intended tasks and contribute towards a more efficacious project team [85]. Therefore, a CPM’s career calling may have a moderating effect on their responsibilities and obligations, which in turn affects their job performance. Thence, the following hypotheses are developed:
**Hypothesis** **4a** **(H4a):**Career calling will moderate the relationship between role conflict and job burnout.
**Hypothesis** **4b** **(H4b):**Career calling will moderate the relationship between role ambiguity and job burnout.
**Hypothesis** **5a** **(H5a):**Career calling will moderate the relationship between role conflict and job performance.
**Hypothesis** **5b** **(H5b):**Career calling will moderate the relationship between role ambiguity and job performance.

### 3.2. Model Specification

A JD-R model is a flexible theoretical framework in which various types of job environments and job characteristics can be classified into the following two categories: job demands and job resources. Job demands and job resources can produce two completely different psychological processes, namely health damage and motivation damage. An over-demanding job will keep individuals in a state of fatigue and lead to emotional exhaustion. A lack of job resources will affect an individual’s job motivation, which will in turn cause the psychological alienation of the individual from their job and reduce their sense of work achievement [20]. Meanwhile, Schaufeli [18] points out that job resources, whether from internal or external factors, can help employees deal with threatening or negative job conditions and promote the completion of job objectives and personal development. Thence, role stress and job burnout belong in the category of job demands, which keep individuals in a state of fatigue and lead to emotional exhaustion. Career calling, on the other hand, as a subjective initiative, is one component of job resources that contributes to alleviating job burnout and role stress. Combined with the high-stress nature of construction projects, this study adopts the JD-R model to conduct theoretical analyses and construct a hypothetical model. These are used to investigate the relationship between role stress, job burnout and job performance in CPMs in China, as shown in Figure 1.

## 4. Method and Data Presentation

### 4.1. Measurements

#### Main Model and Variables

There are three types of sources of the measured items. The first is a direct reference to the measured items in the existing literature (which has proven to have high reliability and validity). The second source is that of modifying the existing items. The last source includes the items that were designed based on the characteristics of China’s construction projects. Face-to-face structured interviews were conducted with industry professionals and researchers, in order to gather their professional comments on the appropriateness of measurements [86]. Next, five experts, who hold positions such as project manager, project engineer and department manager, were selected from different projects to validate the modified questionnaire. Thus, the items of each variable were properly modified according to the nature of the various construction projects and the suggestions of experts [87]. All variables, except for the demographical information survey, were measured using a five-point Likert scale (where 1 means “strongly disagree” and 5 means “strongly agree”). The measurements of each construct are listed in Table 1, along with their standard factor loadings (SFL).

### 4.2. Sampling and Data Collection

This study uses non-probability random sampling, because the respondents were not randomly selected from the population [77,83]. The targeted population includes CPMs in China, because they have a deep understanding of the reality of construction projects and can better answer questions. All the questionnaire surveys were undertaken face-to-face, in order to minimize potential bias. The survey sample was selected from medium- and large-scale construction projects in China. The sample is mainly comprised of owners, contractors, subcontractors and supervisors. Out of the 500 questionnaires sent out, 310 responses were received (response rate = 62.0%), and 191 are considered valid. The other 119 are either incompletely filled out or duplicated from others. In order to avoid non-response bias, this study divided the data into two parts (one part includes non-response data, the other part does not include) and conducted *independent sample t-test*. The results show that the *p* value of each latent variable is greater than 0.05 (*RA*: F = 0.266, *p* = 0.607; *RC*: F = 1.856, *p* = 0.174; *JB*: F = 0.216, *p* = 0.642; *CC*: F = 0.044, *p* = 0.834; *JP*:F = 0.275, *p* = 0.601), indicating that there is no non-response bias. Demographic characteristics of the sample are shown in Table 2.

## 5. Data Analysis and Results

### 5.1. Analysis Strategy

Structural equation modeling (SEM) is a multivariate statistical framework that incorporates regression, factor analysis, and path analysis, which is used to model complex relationships between directly and indirectly latent variables [94]. Compared with multiple regression analysis, SEM makes it more convenient to conduct regression analysis of a model with many dependent variables and perform better in reducing measurement error. More and more studies in construction research use SEM [95,96,97], which reveals the applicability of SEM in this field. Data was analyzed through structural equation modeling (SEM) by using Smart-PLS 3 (Joe F. Hair, University of South Alabama, South Alabama, USA). In general, PLS-SEM is used to measure the relationship between two or more endogenous and exogenous variables [98]. This technique is widely used in the social science studies, because of its ability to simultaneously test multiple dependent and independent variables [99], thereby making it suitable to analyze complex models and small data samples [100]. Single-source data may involve both non-response bias and common method bias [101]. Therefore, the χ^2^ method was used to check non-response bias, and the Harman one-factor test was applied to check the common method bias [97]. The results show a significant heterogeneity between variables and dissipate concerns about the common method bias.

### 5.2. Assessment of Measurement Model

#### 5.2.1. Reliability Testing

The measurement model mainly includes five constructs, namely role ambiguity, role conflict, job burnout, job performance and career calling. In order to test the reliability of different constructs, Cronbach’s alpha values were checked. All alpha values should be greater than the threshold of 0.7 [102]. For this study, the values of Cronbach’s alpha are all greater than 0.70 for all variables. Composite reliability (CR) is another measure of internal consistency. The value of CR should be greater than 0.70, as recommended by Albright [103]. In this study, the value of CR is greater than 0.70, thereby indicating that each construct has good reliability.

#### 5.2.2. Validity Testing

After checking for reliability, the data were checked for validity. The average variance extracted (AVE) has been checked to ensure the convergent validity. The AVE value should be greater than 0.50 [104]. In addition, the data were checked for discriminant validity. The value of the square root of AVE must be greater than the correlations between the construct and those of the other constructs [105]. The reliability and validity results of each construct are shown in Table 3.

### 5.3. Evaluation of Structural Model

The hierarchical regression analysis method was applied to test the research model. The following hierarchical regression analysis model was developed using the bootstrapping method to determine the path coefficients of this paper’s hypotheses. First, due to the fact that certain personal characteristics, such as gender and marital status, may have an impact on job burnout and job performance [106,107], this study examines the effect of these control variables on the dependent variable. The results show that gender and marital status have a non-significant effect on job performance (gender: β = −0.184, *p* > 0.05; marital status: β = 0.053, *p* > 0.05). Besides, considering the effect of sorting of employees (e.g., employees’ wage and work experience) based on their job performance [108], this study supplemented the relevant analysis. The results indicate that CPMs’ wage (β = 0.165, *p* = 0.587) and work experience (β = 0.284, *p* = 0.167) have a non-significant effect on job performance. Because older CPMs may respond differently to perceived stress compared to young CPMs, this study conducted a homogeneity of variance test (role ambiguity (Levene Statistic = 0.428, *p* = 0.733) and role conflict (Levene Statistic = 0.413, *p* = 0.744)); the results shows that the assumption of homogeneity of variance is valid, indicating that older CPMs have the same response to perceived stress compared to young CPMs. Secondly, this study introduces independent variables to analyze the main effect of independent variables on the dependent variable. It was found that H3a (Role Conflict →Job Performance) is non-significant, whereas the remaining hypotheses are significant. Finally, the structural model was used to generate the results for the moderating variable. This study regards career calling as a moderating variable and measures its effect on the relationship between (1) job burnout and job performance and (2) role stress and job performance. This moderating effect was measured by using the “moderation by interaction terms” [99]. The results of hypotheses testing are presented in Table 4 and Figure 2.

To completely understand the effect of moderation by interaction terms, this study has conducted a simple slope analysis. The slopes are presented in Figure 3, Figure 4 and Figure 5, respectively. When lines are not parallel or intersecting, there is a moderating effect. As these figures show, green, red and blue lines separately specify the moderator’s high, mean, and low conditions. Results of the moderation analysis reveal that career calling negatively moderates the relationship between role ambiguity and job burnout, whereas career calling positively moderates the relationship between (1) role conflict and job burnout and (2) role conflict and job performance.

## 6. Discussion

### 6.1. Role Stress and Job Burnout

This study’s findings suggest that role conflict and role ambiguity have negative, significant effects on job burnout (H1a & H1b). In construction projects, there are many cognitive factors affecting role stress in CPMs. Examples of such factors include project deadlines, the number of tasks, and the difficulty of the tasks (e.g., complex decision making, multiple objectives) [43]. Role conflict, as a component of role stress, will have an influence on the emotional experience of personnel (e.g., increased tension, high levels of internal conflict, and decreased job satisfaction) and will increase the sense of job burnout. Besides, role ambiguity will lead to work uncertainty, forcing CPMs to constantly spend their own resources to seek and obtain related information, wasting the time needed to make decisions. Under these circumstances, CPMs will feel physically and mentally exhausted, leading to the formation of job burnout. Therefore, the results of this current study are aligned with previous studies conducted by Kahn [25] and Schaufeli [60], who found that role stress (role conflict and role ambiguity) has a positive effect on job burnout.

### 6.2. Job Burnout and Job Performance

The results support the proposed notion of H2 and show a strong job burnout effect on job performance. The results confirm the findings provided by previous studies, such as Maslach [62] and Wright [68], who found a negative and significant relationship between job burnout and job performance. The symptoms of job burnout not only change the social life of individual attitudes to work, but job burnout also causes disrespect and a distrust of colleagues [109]. Therefore, the presence of specific demands (i.e., role ambiguity and role conflict) and the absence of specific resources (i.e., control coping, social support, and autonomy) predicts burnout, which in turn is expected to lead to various negative outcomes, such as physical illness, turnover, diminished organizational commitment, and a lower level of job performance [110]. Unlike other construction professionals, the role of a CPM requires involvement with the project from its inception to its completion. A CPM’s job performance will have a significant effect on project performance and even an irreplaceable role in a project’s success [23].

### 6.3. Role Stress and Job Performance

This study further posits that role conflict and role ambiguity are related to job performance (H3a & H3b). The results confirm the proposed hypotheses and prove the assumption that role ambiguity has negative effects on the job performance of CPMs. This finding is consistent with previous studies that were undertaken in other contexts [59,60]. However, the results of this study find that role conflict has a non-significant influence on job performance. This finding contradicts prior literature with regard to the role of role conflict in suppressing [69] or improving job performance [76]. According to the viewpoint of Fried [111], role ambiguity and role conflict both have a negative impact on job performance. However, when role ambiguity is at a low level, there is no correlation between role conflict and job performance. In the context of construction projects, a CPM co-ordinates the activities of every project participant. This is done, in order to ensure everyone completes their project tasks within an appropriate time frame, thus contributing to a more efficacious project team. The CPMs also have a greater chance of experiencing role stress. Thence, as two aspects of role stress, role conflict and role ambiguity should be correctly controlled.

### 6.4. The Moderating Role of Career Calling

In this study, career calling has been used as a moderator, in order to find career calling’s moderating effects on the relationships between (1) role stress and job burnout and (2) role stress and job performance. The results reveal that career calling negatively moderates the relationship between role ambiguity and job burnout. Meanwhile, this study also finds that career calling positively moderates the relationship between role conflict and job performance. It is interesting to note that career calling positively moderates the relationship between role conflict and job burnout. However, the moderation effect of career calling on role ambiguity and job performance is insignificant. Therefore, the results of this study provide ample evidence to accept H4a, H4b and H5a, all of which keep pace with the discussions of Duffy [10] and Boyd [85]. As a strong and meaningful passion experienced by individuals in the field of work, career calling is related to life goals and has a pro-social tendency [11], which in turn can positively predict an individual’s self-career clarity, career satisfaction and determination [10]. Career calling contributes to organizational commitment and lower turnover intention. As for the positive moderating effect of career calling on role conflict and job burnout, the dark side of career calling can be better explained. In recent years, scholars have begun to explore the potential dark side of career calling, which lead to some negative effects on individuals [98,99]. Individuals with strong career calling will choose to ignore or even reject other people’s suggestions [112]. This may limit their career development horizons and result in a higher turnover intention [10]. Therefore, an excessive sense of career calling will bring about overload and pressure to individuals. Also, an imbalance of labor and economic returns will lead to job burnout [113]. As safety and environmental matters are essential concerns on a construction site, CPMs are required to consider the safety and environmental aspects of construction projects [114]. This task does require temperate career calling, which is necessary to regulate CPM’s emotions and encourage them to actively participate in their work, especially with the increasing demands, constraints, and complexity within the construction industry.

## 7. Conclusions and Implications

### 7.1. Conclusions

This research attempts to provide some insights into the effect of role stress in CPMs on their job burnout and job performance, according to a JD-R model. Regression analysis results show that: (1) role ambiguity has a negative and significant effect on job burnout and job performance. (2) Role conflict has a negative effect on job burnout but has an insignificant influence on job performance. (3) Job burnout is negatively correlated with job performance. (4) Career calling negatively moderates the relationship between role ambiguity and job burnout and positively moderates the relationship between role conflict and job performance. Interestingly, this study also finds that career calling can positively moderate the effect of role conflict on job burnout, thereby indicating that career calling has the function of a double-edged sword. As such, CPMs should reasonably control the level of career calling and correctly contend with its modulating effect.

### 7.2. Theoretical Implications

Few studies to date have explored the effects of role stress on job burnout and job performance in construction project managers, especially with regard to examining career calling as a moderating variable. Based on the JD-R model, this study investigates the effect of role stress on job burnout and job performance, as well as the moderation effect of career calling. The results are of great theoretical significance to construction project management. Furthermore, this study specifically elaborates on the interactive relationship between job demands and job resources, as well as their effect on job performance.

First, this study contributes to existing role stress literature by specifying two dimensions of role stress (role conflict and role ambiguity). The study also explores the effects of role stress and role ambiguity on job burnout and job performance. In so doing, this study provides an important supplement to existing research in the field of role stress, enriching the study on dimension structure and the measurement tools of role stress, to a certain extent.

Secondly, this study adds knowledge to existing construction management literature, particularly with regard to how to mitigate CPM role stress and job burnout. This study not only considers organizational support, but it also takes into account the power of self-efficacy, such as career calling. Therefore, this study regards career calling as a moderating variable, probing into the regulatory mechanism formed by career calling and helping us to better understand how to use career calling to exhibit its influence.

### 7.3. Practical Implications

This current research sheds some light for practitioners, especially for project stakeholders, on how to better control their CPMs’ job burnout and enhance job performance. Firstly, to strengthen the role stress management of CPMs, construction enterprises should establish and improve their role stress monitoring and prevention management systems. Specifically, construction enterprises should actively introduce an EPA program (employee assistance program) [115] and regularly implement stress assessment. Counseling should also be available, to train CPMs on how to adapt to stress. Specifically, professional psychological counseling services should be provided to CPMs, and long-term stress warning and control mechanisms should be formed. 

Secondly, CPMs should receive clear role orientation, and clear job objectives, responsibilities, authority, and expectations for jobs should be set. These steps could help role ambiguity and conflict. Thirdly, the organizational structure should be flattened, and the communication between middle managers, senior leaders and colleagues should be strengthened, in order to reduce unnecessary paradoxical information.

Thirdly, the organizational climate is CPMs’s overall perception of the enterprise, which can have an important impact on their understanding of organizational life [116]. Therefore, construction enterprises should strive to create a supportive atmosphere. Specifically, enterprises should respect and attach importance to the needs of CPMs, appropriately encourage them to put forward their own opinions and suggestions on the company’s operation, strive to meet their reasonable requirements, and actively encourage CPMs innovation. At the same time, construction enterprises should improve the working environment, provide abundant training opportunities for CPMs, and increase the reasonable scope of authorization, which will help reduce the job stress.

Finally, although career calling can bring about more work engagement, blindly emphasizing career calling will cause CPMs to passively work overtime passively. This will disturb CPMs’ input in other important roles and then have a negative impact on their personal life, physical and mental health, and job performance. Therefore, construction enterprises should pay closer attention to the potential negative impact of career calling on the personal life and occupational health of CPMs. Enterprises should actively improve the welfare of CPMs, encouraging them to maintain a work-life balance, thus constantly improving their life quality, enhancing the happiness of CPMs, and promoting the sustainable development of human resources.

### 7.4. Limitations and Future Work

This research explores ways to improve the job performance of CPMs, by exploring the relationships between role stress, job burnout, and career calling. Notwithstanding the foregoing contributions and implications to academia and practice, this study has several limitations that require future research. Firstly, the sample data are limited to specific regions of China. The recommended future direction would be to collect data from different countries or regions, in order to explore the relationships of role stress, job burnout, career calling, and job performance from different cultural perspectives. Secondly, this study only takes job burnout as a mediating variable to explore the impact of role stress on job performance. Future research can consider and discuss other variables, such as work–family conflict. Thirdly, Due to the limitation of energy and abilities, this study only considers the moderating role of career calling. Future research can add organizational support, so as to investigate the impact of the interaction between organizational support and career calling on job burnout and job performance. Finally, this study ignored the impact of personality traits, such as the big-five personality traits, on research results. Therefore, future work can add personality traits into study and investigate the different responses to role stress and job burnout under the influence of personality traits.

## Figures and Tables

**Figure 1 ijerph-16-02394-f001:**
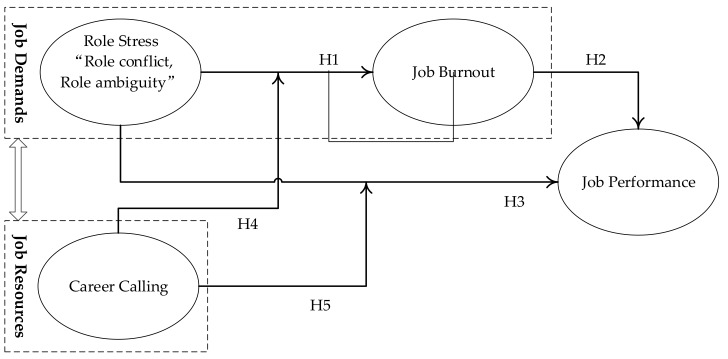
Theoretical model underlying empirical research.

**Figure 2 ijerph-16-02394-f002:**
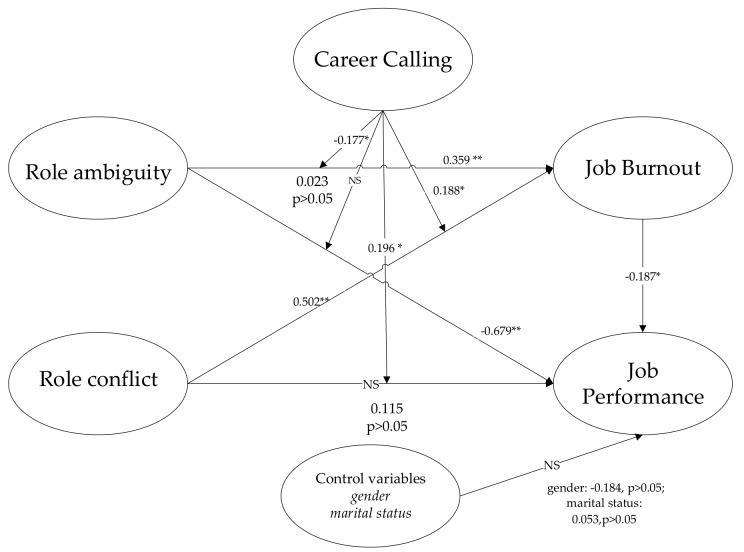
Results of model and hypothesis testing. Note: * Significant at *p* < 0.05; ** Significant at *p* < 0.01; NS indicates not supported.

**Figure 3 ijerph-16-02394-f003:**
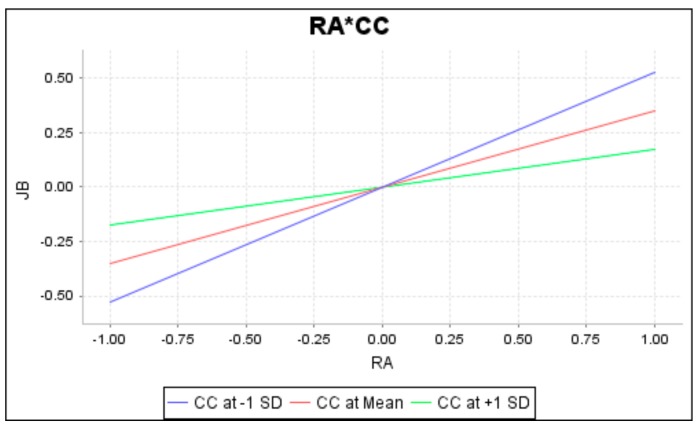
Moderation of career calling on the relationship between role ambiguity and job burnout.

**Figure 4 ijerph-16-02394-f004:**
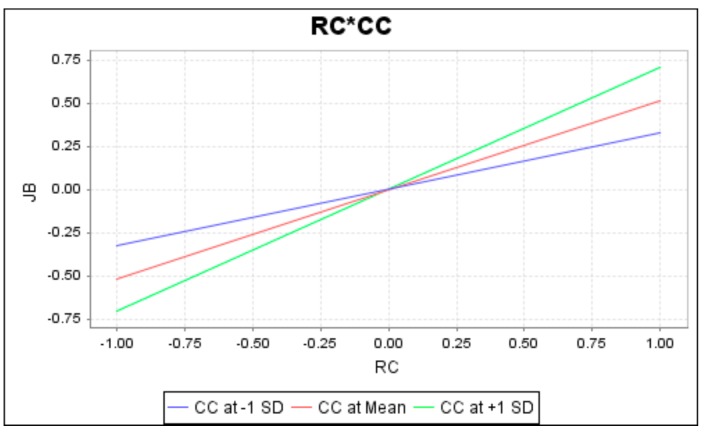
Moderation of career calling on the relationship between role conflict and job burnout.

**Figure 5 ijerph-16-02394-f005:**
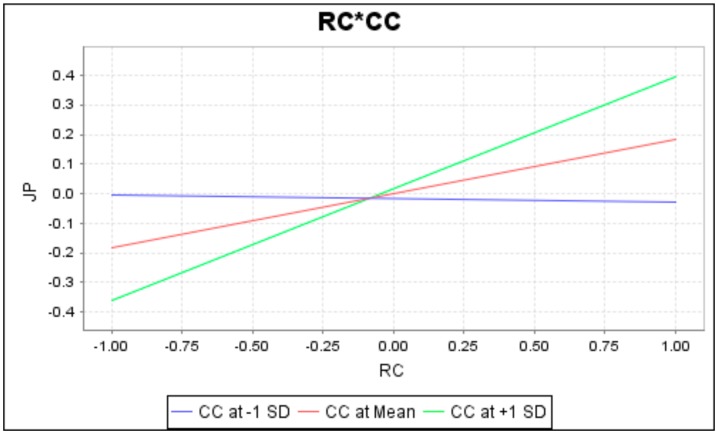
Moderation of career calling on the relationship between role conflict and job performance.

**Table 1 ijerph-16-02394-t001:** Measurements for variables.

Variables	No.	Measurement	SFL	References
Role Ambiguity (RA)	RA1	I feel uncertain about how much authority I have in my job.	0.933	[17,71]
	RA2	I don’t have clear, planned goals and objectives for my job.	0.963	
	RA3	I don’t know exactly what is expected of me in my job.	0.926	
	RA4	I don’t receive clear explanations of what has to be done in my job.	0.935	
Role Conflict (RC)	RC1	I receive incompatible requests from two or more people.	0.934	
	RC2	I do things that are apt to be accepted by one and not by others.	0.934	
	RC3	I receive assignments without adequate resources to execute them.	0.929	
	RC4	I work on unnecessary things.	0.897	
Job Burnout (JB)	JB1	Work makes me feel physically and mentally exhausted.	0.789	[3,88,89]
	JB2	I feel used up after work.	0.889	
	JB3	I feel tired when I wake up and have to face the day’s work.	0.903	
	JB4	Working all day is really stressful for me.	0.870	
	JB5	Work makes me feel like I’m going to collapse.	0.818	
	JB6	I have become less and less interested in my work since I started this job.	0.820	
	JB7	I am not as enthusiastic about my work and colleagues as I used to be.	0.821	
	JB8	I doubt the meaning of my work.	0.885	
Job Performance (JP)	JP1	Carried out the core parts of my job well.	0.874	[90,91]
	JP2	Completed my core tasks well using the standard procedures.	0.870	
	JP3	Ensured my tasks were completed properly.	0.887	
	JP4	Adapted well to changes in core tasks.	0.887	
	JP5	Coped with changes to the way you have to do your core tasks.	0.881	
	JP6	Learned new skills to help you adapt to changes in your core tasks.	0.919	
	JP7	I am willing to make my efforts for job goals.	0.895	
	JP8	I put a lot of energy into my job.	0.905	
	JP9	I won’t give up my job easily.	0.861	
Career Calling (CC)	CC1	I feel that I am destined to do my current job.	0.802	[54,92,93]
	CC2	Engaging in my current job makes me feel the meaning of life.	0.848	
	CC3	Compared with other jobs, I deserve to be in my present job.	0.920	
	CC4	The value of my life depends largely on the job I do.	0.888	
	CC5	My job is beneficial to others.	0.725	
	CC6	My job can meet the needs of society and contribute to society.	0.888	

**Note: SFL** indicates standard factor loadings.

**Table 2 ijerph-16-02394-t002:** Demographic characteristics of the sample (N = 191).

Demographic	Value	Frequency	%
Working experience	5–10 years	45	23.6
	10–15 years	80	41.9
	>15 years	66	34.6
Age	25–30 years	27	14.1
	31–35 years	52	27.2
	36–45 years	65	34.0
	>45 years	47	24.6
Project party	Owners	49	25.7
	Contractors	60	31.4
	Subcontractors	52	27.2
	Supervisions	30	15.7
Educational background	Junior college and below	57	29.8
	Bachelor	82	42.9
	Master	30	15.7
	Doctor	22	11.5
CPMs’ wage	3000–5000 yuan	22	11.5
	5001–7000 yuan	76	39.8
	7001–9000 yuan	69	36.1
	>9000 yuan	24	12.6

**Table 3 ijerph-16-02394-t003:** The reliability and validity of constructs.

Variables	CR	Cronbach’s Alpha	AVE	RA	RC	JB	JP	CC
RA	0.968	0.955	0.882	0.939	-	-	-	-
RC	0.959	0.943	0.854	0.707	0.924	-	-	-
JB	0.954	0.945	0.723	0.714	0.756	0.850	-	-
JP	0.971	0.966	0.786	−0.731	−0.506	−0.585	0.887	-
CC	0.938	0.920	0.719	−0.299	−0.418	−0.327	0.238	0.848

Note: RA, role ambiguity. RC, role conflict. JB, job burnout. JP, job performance. CC, career calling. Square roots of AVE are presented in bold on the diagonal.

**Table 4 ijerph-16-02394-t004:** Hypotheses decision table.

Hypothesis	Path Coefficient	T Statistics	*p* Values	Effect Size f 2	Hypotheses Decision
JB→JP	−0.187	2.185	0.029	0.028	H2a: Supported
RC→JB	0.502	5.845	0.000	0.346	H1a: Supported
RC→JP	0.115	1.335	0.182	0.011	H3a: Not Supported
RC*CC→JB	0.188	2.457	0.014	0.044	H4a: Supported
RC*CC→JP	0.196	2.446	0.015	0.040	H5a: Supported
RA→JB	0.359	4.228	0.000	0.177	H1b: Supported
RA→JP	−0.679	9.387	0.000	0.432	H3b: Supported
RA*CC→JB	−0.177	2.243	0.025	0.039	H4b: Supported
RA*CC→JP	0.023	0.301	0.763	0.001	H5b: Not Supported

Model fit: SRMR = 0.042; χ2/*df* = 3.257; NFI = 0.905.

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
