# Peer review of "Role Stress, Job Burnout, and Job Performance in Construction Project Managers: The Moderating Role of Career Calling"

_ijerph, 2019, doi:10.3390/ijerph16132394_

Round 1

Reviewer 1 Report

Comments

1.The focus on the Chinese context should be motivated in the revised introduction.

2.The response rate of the survey was 62% (page 9). Was non-response to the survey random or not? This information would be useful in order to better understand the estimation results that are reported in the paper.

3.Do the data contain (survey) weights or not? 

4.There is an important issue related to the interpretation of the estimation results. Employees are not randomly assigned into workplaces. Failure to account for sorting of employees will bias any estimated effects. The size of the bias is not known. This problem can be addressed using information on employees’ wage and work histories (Bockerman et al. 2012). This issue should be discussed in the revised version.

5.The estimates may suffer from the omitted-variable bias (see also point 4 above). For example, personality traits are most likely correlated with all (subjective) measures that are used in the analyses. This issue should also be noted in the revised version. 

6.The paper does not consider the heterogeneity in the estimated effects. The relationships can differ significantly e.g. by age. Older workers may respond differently to perceived stress compared to young workers.

7.The concluding section of the paper should state more avenues for future studies. 

Reference

Bockerman, P., Bryson, A. & Ilmakunnas, P. (2012).Does high involvement management improve worker wellbeing?Journal of Economic Behavior and Organization, 84:6, 660-680.

Author Response

A1: The focus on the Chinese context should be motivated in the revised introduction.

Response: Thanks for your comments. We have added the Chinese context to discuss this study at Line 32-41. The details as follows:

The Chinese construction industry has been expanding at a high growth rate over the past 20 years. In order to alleviate the effects of the global financial crisis, the Chinese government has vowed to increase investments in housing and infrastructural works to stimulate the national economy [2]. In 2017, China's construction industry grew rapidly, with its total output value reaching 21,395 billion yuan, an increase of 10.5% over the previous year, much higher than 7.1% in 2016 and 2.3% in 2015. This is equivalent to 25.9% of China's GDP, while the total added value accounts for 6.7% of GDP. The contribution of construction industry to China's economy is also evident. China's construction contractors are undertaking a large number of domestic and foreign construction projects, bringing huge occupational pressure and psychological burden to CPMs [3].

A2: The response rate of the survey was 62% (page 9). Was non-response to the survey random or not? This information would be useful in order to better understand the estimation results that are reported in the paper.

Response: Thank you for your comments. The non-response bias would be useful in order to better understand the estimation results that are reported in the paper, we supplement the relevant analysis at Line 343-348. The details as follows:

In order to avoid non-response bias, this study divided the data into two parts (one part includes non-response data, the other part does not include) and conducted independent sample T-test. The results show that the P value of each latent variable is greater than 0.05 (RA: F=0.266, P=0.607; RC: F=1.856, P=0.174; JB: F=0.216,P=0.642; CC: F=0.044P=0.834JP: F=0.275, P=0.601), indicating that there is no non-response bias.

A3: Do the data contain (survey) weights or not?

Response: Thank you for your comments. Firstly, this study is not a national large-scale comprehensive survey, and does not need to reflect the macro-level of research issues through weights of data. Secondly, this study uses mature measurement items, which can reflect the potential concept, without giving weight to different dimensions after a series of analysis. Moreover, SEM (structural equation modeling) method is able to reflect the potential construct, without using the mean method to represent and form corresponding constructs.

A4: There is an important issue related to the interpretation of the estimation results. Employees are not randomly assigned into workplaces. Failure to account for sorting of employees will bias any estimated effects. The size of the bias is not known. This problem can be addressed using information on employees’ wage and work histories. This issue should be discussed in the revised version.

Response: Thanks for your advice. Considering the simplicity of the article, we have done a survey on employees’ wage and work histories before, but it is not shown in the article. According to your suggestion, we have supplemented the relevant analysis at Line 391-394 and in Table 2. The details as follows:

Besides, considering the effect of sorting of employees (eg., employees’ wage and work experience) on their job performance [110] , this study supplemented the relevant analysis. The results indicate that CPMs’ wage (β=0.165, P=0.587) and work experience (β=0.284, P=0.167) have a non-significant effect on job performance.

Table 2. Demographic characteristics of the sample (N=191).

Demographic

Value

Frequency

%

Working experience

5-10 years

45

23.6

10-15 years

80

41.9

> 15 years

66

34.6

Age

25-30 years

27

14.1

31-35 years

52

27.2

36-45 years

65

34.0

> 45 years

47

24.6

Project party

Owners

49

25.7

Contractors

60

31.4

Subcontractors

52

27.2

Supervisions

30

15.7

Educational background

Junior college and below

57

29.8

Bachelor

82

42.9

Master

30

15.7

Doctor

22

11.5

CPMs’ wage

3,000-5,000 yuan

22

11.5

5,001-7,000 yuan

76

39.8

7,001-9,000 yuan

69

36.1

> 9,000 yuan

24

12.6

A5: .The estimates may suffer from the omitted-variable bias (see also point 4 above). For example, personality traits are most likely correlated with all (subjective) measures that are used in the analyses. This issue should also be noted in the revised version.

Response: Thank you for your comments. We feel very sorry that this issue cannot be solved for the time being, because our questionnaires were distributed several months ago, when we ignored the impact of personality traits on our results. And it is not realistic to recollect these data now. Inspired by your advice, we add this shortcoming into the section of limitations and future work at Line 565-568 and investigate this issue in future studies. The details as follows:

Finally, this study ignored the impact of personality traits, such as big-five personality on research results. Therefore, future work can add personality traits into study and investigate the different responses to role stress and job burnout under the influence of personality traits.

A6: The paper does not consider the heterogeneity in the estimated effects. The relationships can differ significantly e.g. by age. Older workers may respond differently to perceived stress compared to young workers.

Response: Thanks for your comments. Considering the heterogeneity in the estimated effects, we have added the homogeneity of variance test at Line 395-399. The details as follows:

Because older CPMs may respond differently to perceived stress compared to young CPMs, this study conducted a homogeneity of variance test, role ambiguity [role ambiguity (Levene Statistic=0.428, P=0.733) and role conflict (Levene Statistic=0.413, P=0.744)], the results shows that the assumption of homogeneity of variance is valid, indicating older CPMs have the same response to perceived stress compared to young CPMs.

A7: The concluding section of the paper should state more avenues for future studies.

Response: Thanks for your advice. We have add some avenues for future studies at Line 562-565. The details as follows:

Thirdly, Due to the limitation of energy and abilities, this study only considers the moderating role of career calling. Future research can add organizational support, so as to investigate the impact of the interaction between organizational support and career calling on job burnout and job performance.

Reviewer 2 Report

Dear authors,

The paper presented contribute to the understanding of what factors could moderate the relationship between different job stressors in the context of CPMs in the Chinese environment

From my perspective, the article would be interesting for scholars interested in management the main stressors in CPM.

Although this study shows a potential value, I suggest authors to incorporate the recommendations indicated in the reviewed version attached to this report.

In brief this are the main concerns that I observe after reading the paper:

Reduce the extension of the introduction section.

Include a figure to explain the nomological network that links the variables under study.

Explain and better justifiy the selection of SEM models as data analysis strategy pertinent to the research objectives.

Incorporate some missing SEM coefficients that should been reported in this kind of analysis.

Suggest policy implications of this research (for example to inform the design of organizational interventions to improve the levels of stress)

Correct references and some citations mistakes within the text.

Best regards,

---

Author Response

Reviewer 2

B1: Reduce the extension of the introduction section.

Response: Thanks for your comments. Considering the simplicity of the article, we have deleted the research objects at Line 69-72. The details as follows:

The specific objectives of this study are: i) exploring the influence of role stress on job burnout and job performance; ii) exploring the effects of job burnout on job performance; and iii) investigating the moderating effect of career calling on job burnout and job performance.

B2: Include a figure to explain the nomological network that links the variables under study.

Response: Thank you for your comments. In order to show the logic of this study more clearly, we revised the theoretical framework. As shown in Figure 1, role stress (role conflict and role ambiguity) and job burnout belong in the category of job demands, which keep individuals in a state of fatigue and lead to emotional exhaustion. Career calling, on the other hand, as a subjective initiative, is one component of job resources that contributes to alleviating job burnout and role stress. Thence, role stress (role conflict and role ambiguity) may have a negative effect on job burnout and job performance (H1-H3). Career calling not only reduces individual role stress, job burnout, and maintain mental health; it also improves an employee’s work meaning to obtain life satisfaction and experience happiness, which indicates that career calling may have a moderating effect on CPMs responsibilities and obligations, which in turn affects job burnout (H4) job performance (H5).

Figure 1. Theoretical model underlying empirical research.

B3: Explain and better justify the selection of SEM models as data analysis strategy pertinent to the research objectives.

Response: Thanks for your advice. We have explained and better justified the selection of SEM models as data analysis strategy at Line 352-357. The details as follows:

Structural equation modeling (SEM) is a multivariate statistical framework that incorporates regression, factor analysis, and path analysis, which is used to model complex relationships between directly and indirectly latent variables [97]. Compared with multiple regression analysis, SEM makes it more convenient to conduct regression analysis of a model with many dependent variables and perform better in reducing measurement error. More and more studies in construction research use SEM [98,99], which reveals the applicability of SEM in this field.

B4: Incorporate some missing SEM coefficients that should been reported in this kind of analysis.

Response: Thanks for your advice. We have shown some missing SEM coefficients at Line 390-391 and in Table 4 and Fig. 2. The details as follows:

The results show that gender and marital status have a non-significant effect on job performance (gender: β=-0.184, P>0.05; marital status: β=0.053, P>0.05).

Table 4. Hypotheses decision table.

Hypothesis

Path   coefficient

T   Statistics

P   values

Effect   size f 2

Hypotheses   decision

JB→JP

-0.187

2.185

0.029

0.028

H2: Supported

RC→JB

0.502

5.845

0.000

0.346

 H1a:   Supported

RC→JP

0.115

1.335

0.182

0.011

     H3a:   Not Supported

RC*CC→JB

0.188

2.457

0.014

0.044

H4a: Supported

RC*CC→JP

0.196

2.446

0.015

0.040

H5a: Supported

RA→JB

0.359

4.228

0.000

0.177

H1b: Supported

RA→JP

-0.679

9.387

0.000

0.432

H3b: Supported

RA*CC→JB

-0.177

2.243

0.025

0.039

H4b: Supported

RA*CC→ JP

0.023

0.301

0.763

0.001

    H5b:   Not Supported

Model fit: SRMR=0.042; χ2/df=3.257; NFI=0.905

Fig. 2. Results of model and hypothesis testing

B5: Suggest policy implications of this research (for example to inform the design of organizational interventions to improve the levels of stress).

Response: Thanks for your advice. We have added policy implications of this research at Line 537-544. The details as follows:

 Thirdlyorganizational climate is CPMs's overall perception of the enterprise, which can have an important impact on their understanding of organizational life [120]. Therefore, construction enterprises should strive to create a supportive atmosphere. Specifically, enterprises should respect and attach importance to the needs of CPMs, appropriately encourage them to put forward their own opinions and suggestions on the company's operation, strive to meet their reasonable requirements, and actively encourage CPMs innovation. At the same time, construction enterprises should improve the working environment, provide abundant training opportunities for CPMs, and increase the reasonable scope of authorization, which will help reduce the job stress.

B6: Correct references and some citations mistakes within the text.

Response: Thanks for your advice. We have corrected references and some citations mistakes within the text at Line 141-142. The details as follows:

She proposes that job burnout can be analyzed from three dimensions: emotional exhaustion, cynicism and low professional efficacy.

B7: Please add information about the sample and the kind of organization(s) evaluated.

Response: Thanks for your advice. We have added relevant information about the sample and the kind of organizations at Line 17-18 in Abstract and elaborated the data collection process in detail at Line 339-343. The details as follows:

Using data from 191 owners, contractors, subcontractors and supervisors in the Chinese construction industry.

The survey sample was selected from medium- and large-scale construction projects in China. The sample is mainly comprised of owners, contractors, subcontractors and supervisors. Out of the 500 questionnaires sent out, 310 responses were received (response rate = 62.0%), and 191 are considered valid. The other 119 are either incompletely filled out or duplicated from others.

B8: Please mention some official statistics referred to the context under study.

Response: Thank you for your advice. We have added some official statistics referred to the context under study at Line 35-39. The details as follows:

In 2017, China's construction industry grew rapidly, with its total output value reaching 21,395 billion yuan, an increase of 10.5% over the previous year, much higher than 7.1% in 2016 and 2.3% in 2015. This is equivalent to 25.9% of China's GDP, while the total added value accounts for 6.7% of GDP. The contribution of construction industry to China's economy is also evident.

B8: Are (organizational support and social support) the same? Please offer an explanation.

Response: Thank you for your comments. There is a certain difference between organizational support and social support. Eisenberger et al [1] defined organizational support as the overall perception and belief of employees about how an organization views their contributions and cares about their interests. McMillin [2] divided organizational support into instrumental support and social emotional support. Therefore, the organizational support of the construction industry can be defined as the attitude of the construction company toward employee’s contributions, as well as the concern of the construction company for the welfare of its employees. Different from organizational support, social support is mainly divided into into supervisor support and coworker support[3]. Supervisor support encompasses, for instance, caring about subordinates, valuing their contributions, helping them on work-related issues, and facilitating their skill development[4]. Coworker support refers to the degree of assistance enacted by work colleagues[5]. The support from coworkers includes the provision of caring, tangible aid, and information[6].

B9: Please offer a standard definition of both constructs (role conflict and role ambiguity).

Response: Thank you for your comments. We have offered a standard definition of both constructs (role conflict and role ambiguity) in manuscript at Line 123-127. The details as follows:

Role conflict means that, when individuals are faced with two or more role expectations, they cannot meet both of these expectations at the same time [36]. Role ambiguity, on the other hand, refers to employees’ feelings when they are unclear or lack proper understanding of their role and cannot obtain clear role expectations at work [37].

B10: Many studies suggest that the roles should to be incompatible between them.

Response: Thank you for your comments. Role conflict mainly refers to the conflict between the organization's norms and evaluation standards and individual role requirements, or the conflict between multiple role expectations in an organization[7]. Meanwhile, role conflicts are mainly caused by the inconsistency between role norms and behaviors[8]. From the definition of role conflict by different scholars, it can be seen that although there is no unified definition of this concept, it is generally agreed that role conflict mainly refers to the psychological and physical discomfort caused by the individual's inability to meet the requirements of different roles at the same time when facing various role requirements.

In addition, role conflict does exist in the construction industry, and the purpose of this study is not to achieve role compatibility, but to understand the role conflict mechanism and propose effective countermeasures to better improve job performance and project performance while maintaining the nature of role conflict.

After-note: the authors hope that they have now addressed all the concerns, feedback and suggestions of the Editor and Reviewers adequately. Nevertheless, if the Editor and Reviewers have any additional comments, the authors would be pleased to address these further. Thank you again for your positive contributions to make this a better paper.

Round 2

Reviewer 2 Report

Dear authors,

Thank you so much for introduce the corrections in the last version of the manuscript.

After reading your response to my review report I appreciate your efforts for improving the quality of the paper.

I consider the paper could be published in the present form.

Best regards,

---